# An Exploratory Comparison of Alpha and Beta Network Connectivity Across Four Depression Subtypes

**DOI:** 10.3390/jcm14155295

**Published:** 2025-07-26

**Authors:** Christopher F. Sharpley, Ian D. Evans, Vicki Bitsika, Kirstan A. Vessey, G. Lorenzo Odierna, Emmanuel Jesulola, Linda L. Agnew

**Affiliations:** 1Brain-Behaviour Research Group, School of Science & Technology, University of New England, Queen Elizabeth Drive, Armidale, NSW 2351, Australia; ievans3@une.edu.au (I.D.E.); kvessey@une.edu.au (K.A.V.);; 2Department of Neurosurgery, The Alfred Hospital, Melbourne, VIC 3004, Australia; 3School of Health, Griffith University, Brisbane, QLD 4222, Australia

**Keywords:** depression, subtypes, EEG, connectivity, alpha, beta

## Abstract

**Background/Objectives:** Depression is a major disorder that has been described in terms of its underlying neurological characteristics, often measured via EEG. However, almost all previous research into the EEG correlates of depression has used a unitary model of Major Depressive Disorder (MDD), whereas there is strong evidence that MDD is heterogeneous in its symptomatology and neurological underpinnings. **Methods:** To investigate the EEG signatures of four subtypes of depression defined according to the previous literature, the Zung Self-rating Depression Scale was administered to 54 male and 46 female volunteers (M age = 32.53 yr). EEG data were collected during an Eyes Closed condition and examined for differences in connectivity across brain networks in the alpha- and beta-bands. **Results:** The results were examined in terms of the number and direction of connectivity differences between depressed and non-depressed participants within each depression subtype, the alpha- and beta-band connectivities, the regions of the brain that were connected, and the possible functional reasons why specific brain regions were differently connected for depressed and non-depressed participants within each MDD subtype. **Conclusions:** The results suggested some differences in the alpha- and beta-band connectivity between some of the MDD subtypes that are worth considering as representing different neurological signatures across the depression subtypes. These findings represent an initial challenge to defining depression as a unitary phenomenon, and suggest possible benefits for further research into the underlying neurological phenomena of depression subtypes.

## 1. Introduction

### 1.1. Major Depressive Disorder

Although often considered to be a unitary disorder, Major Depressive Disorder (MDD) is defined in terms of nine heterogeneous major diagnostic criteria [1] which, when combined with the Associated Features of MDD, produce almost 1500 distinct clusters of these symptoms that are sufficient to meet the diagnosis of MDD [2]. This acknowledged heterogeneity of MDD [3] argues for diagnostic processes that are based on individualized symptom profiles, which may hopefully lead to more targeted and effective treatments [4]. One pathway to the more individualized diagnosis of MDD is via consideration of depression subtypes [5], which may represent a more valuable model for the diagnosis and treatment of depression than considering it as a unitary disorder [5,6,7]. Previous research has reported on several models of MDD subtypes, including melancholia, psychotic depression, atypical depression, and anxious depression [8], or depressed mood, anhedonic depression, cognitive depression, and somatic depression [9,10]. The latter model of MDD subtypes is based upon the presence of clinically cohesive subgroups of MDD symptoms, plus their theoretical neurobiological underpinning pathways, and has been verified within three patient populations. This MDD subtype model was investigated in this study.

### 1.2. Neurobiology of MDD Subtypes

The identification of the neurobiological underpinnings of subtypes of MDD would assist in the development of a comprehensive model of these depression subtypes, and also feed into more detailed assessment, diagnosis, and targeted treatment options [11,12,13]. To contribute to that process, the investigation of brain site connectivity patterns in different MDD subtypes represents a potentially valuable step in building the validity of MDD subtype models, as well as contributing to the accumulation of possible ‘causal’ neurobiological pathways of depressive behaviour [14]. Several imaging techniques have been applied to the task of distinguishing MDD subtypes [15]. For example, Wang et al. (2021) used fMRI data to identify an insomnia-dominated subtype, and another MDD subtype that was characterized by prominent anhedonia and hypoconnectivity in the subcortical and dorsal attention networks [16]. Toenders et al. (2020) were able to identify three MDD subtypes using structural MRI data on cortical surface area and thickness (severe depression with increased appetite, severe depression with decreased appetite and severe insomnia, and moderate depression) [17]. Importantly, these studies used an *a posteriori* model of analysis, by collecting data on neurological phenomena and then regressing those data to identify specific subgroups of MDD symptoms. That is a legitimate approach to the problem of describing the neurological underpinning of MDD subtypes, but it is not the only valid approach to that goal. In the clinical setting, patients present with various clusters of MDD symptoms, and the clinician needs to be able to work with that presentation. Thus, an alternate model of exploring the neurological bases of MDD subtypes is to define the subtype according to a rational theoretical grouping of MDD symptoms (as in the DSM-5-TR and ICD-11) and then exploring any neurological differences between those groups of MDD symptoms. Because the identification of neurological bases of MDD subtypes has been described as being in “an incipient exploratory stage” [18], with suggestions about the use of various methodologies, these two approaches are equally valuable. In this study, the second method was followed, using the pre-defined four subtypes of MDD described above (i.e., depressed mood, anhedonic depression, cognitive depression, and somatic depression [9,10]). This approach was adopted because it is based upon subgroups of MDD symptoms that may be argued to cohere on the basis of the associations between the symptoms themselves, and is therefore directly relevant to the everyday clinical management of MDD-presenting patients.

### 1.3. EEG and MDD

A great deal of research into the neurophysiological substrates of depression has been undertaken via EEG measurements of brain activity. Although there are several different frequencies of brain waves that may be used to infer different levels of activation in specific brain regions, two which are of direct relevance to how the individual’s brain is processing information are alpha (8 Hz to 12.9 Hz), generally indicative of a relaxed mental state [19], and beta (13 Hz to 18 Hz), which represents more concentrated electrical activity, usually associated with problem-solving or more stressful conditions [20,21]. Limitations in an individual’s ability to maintain a relaxed mental state despite stressors, or to focus appropriate levels of problem-solving on the major issues they face, are key aspects of the diagnostic symptomatology of MDD [1], and so have a *prima facie* role in the presence and severity of that disorder. If these states were identified within specific subtypes of MDD, then the role of neurological substrates in MDD might be clarified.

### 1.4. Brain Connectivity and MDD

As described by Fox (2018) and confirmed by Wang et al. (2021), one potentially valuable index of the neurological processes underlying MDD subtypes is the connectivity of brain regions [22,23]. In particular, differences between connectivity patterns of depressed versus non-depressed persons could provide direct indications of how a particular subtype of MDD influenced the efficiency of brain regional network communications. Data regarding this connectivity can be of value because it provides intricate information about local and global neural processing as an interactional process rather than about the brain when considered as isolated regions, thereby more realistically reflecting the ‘network’ nature of brain functions [24], which can help explain behaviour, thinking, and emotions [25], and also symptoms of psychiatric disorders [22].

Clearly, brain network connectivity data have the potential to increase understanding of the ways that neurological phenomena are associated with depression subtypes. Several recent reports on this process have suggested the possible presence of a ‘depression’ circuit [26,27,28], but for a global depression metric rather than depression subtypes. Additionally, some attention has been given to specific groups of MDD symptoms, such as those associated with psychomotor disturbance [29], anhedonia [30], sadness [31], and cognitive difficulties [32]. There have been some suggestions of developing neural models of depression biotypes to fit these particular symptom groupings [33], some of which have approximated the four depression subtypes identified above (i.e., Depressed mood, Anhedonia, Cognitive depression, Somatic depression) (e.g., Hack et al., 2023; Nakajima et al., 2022; Wüthrich et al., 2024; T. Zhang et al., 2021) [29,32,34,35]. At this stage, those investigations have yet to report brain network connectivity data specific to the four groups of symptoms represented by those MDD subtypes.

Brain network connectivity may be defined as structural (anatomical connections between brain regions), effective (how one brain region influences others), and functional connectivity; the latter is represented by statistical associations between signals from different brain regions that may be used to determine patterns of activity between those regions [36]. These data are valuable to the exploration of neurological processes underlying psychiatric disorders because specific band (e.g., alpha band, beta band, etc.) functional connectivity information has been shown to relate to white matter defects which are associated with those disorders [37,38].

As an indicator of the large amount of research on the association between functional connectivity and depression, Miljevic et al. (2023) reviewed 52 studies, 36 of which examined resting state EEG data [24]. These studies were sufficiently heterogeneous in methodology to prevent definite conclusions being drawn regarding the associations between functional connectivity and MDD, leading Miljevic et al. (2023) to comment that, although there were some “reasonably consistent suggestions” ((p. 298) [24]) that depressed persons exhibited higher alpha functional connectivity at frontal sites, there were too many inconsistencies across other brain regions and frequency bands to draw any firm conclusions. Those authors concluded that further research was needed to detect associations between depression and other frequency bands besides alpha. They also cautioned against adopting an overly optimistic approach that assumed that different clear-cut conclusions could be drawn regarding connectivity and MDD in general. Instead, the expectation of a high degree of overlap between aspects of MDD and different brain region connectivities was urged as a more realistic approach to the complexity of brain wave activity and the heterogeneity of MDD.

### 1.5. Diagnosing MDD

One of the major methodological limitations of most of the previous research reviewed by Miljevic et al. (2023) was how MDD or “depression” was diagnosed [24]. While some studies applied clinical interviews or self-report inventories based on standardized diagnostic criteria for MDD [1,39], others did not report their diagnostic processes. More relevantly for the precise application of personalized medicine models of diagnosis and treatment [4,40,41], none of the 52 studies reviewed by Miljevic et al. (2023) used diagnostic methodologies that included subtypes of depression, or even the less-definite symptom profiles of MDD symptomatology that have been suggested as increasing the likelihood of more effective treatment outcomes [24,42,43,44,45].

### 1.6. Aims of the Study

Therefore, this study aimed to identify differences between the alpha- and beta-wave connectivity patterns of four clinically defined MDD subtypes: Depressed mood, Anhedonia, Cognitive depression, and Somatic depression [9]. The primary focus was on whether the scores on these four MDD subtypes could be associated with different incidences of connectivity (i.e., the number of connections) between brain regions, or different patterns of connectivity (i.e., which regions were involved) at alpha and beta band frequencies. Due to the lack of previous studies of these four MDD subtypes and their comparative EEG data, it was not possible to set this study within the previous research. Instead, this study is necessarily exploratory, seeking to identify possible associations that might be used as hypotheses for future research. Data were collected via EEG from community volunteers because this study is part of a larger research project that is focused upon correlates of depression and anxiety in the general community. Electrical source analysis was used to depict the connectivity patterns for alpha- and beta-wave activity, and to identify any differences in those connectivity patterns between depressed and non-depressed participants. The method of defining MDD subtypes by their (homogeneous) clinical symptoms and then exploring any differences in brain connectivity was adopted because of its closer resemblance to clinical practice. A major caveat was adopted that the expectation of finding clear-cut differences in the associations between four MDD subtypes and two types of electrical activity in the brain may have been overly simplistic. MDD is heterogeneous, and alpha and beta waves are reasonably distinct phenomenon, but the likelihood of identifying statistically significant and exclusive differences between the four MDD subtypes in terms of their alpha- and beta-wave connectivities may be strongly diminished in a system of such complexity as is represented by the multiple MDD symptoms represented in Table 1 and two forms of brain electrical activity.

## 2. Methods

### 2.1. Participants

Participants were 100 adult volunteers (54 males, 46 females) aged between 18 yr and 75 years (*M* age = 32.53 yr, SD = 14.13 yr), recruited from the New England region of New South Wales, Australia for a larger study [46]. Selection criteria included: no previous medical history of severe physical brain injury, brain surgery, or history of epilepsy or seizure disorder, or claustrophobia (EEG data were collected in a small booth). Although some EEG studies have selected participants on the basis of handedness, that was not done here because 61% to 70% of left-handed people also have left hemispheric dominance [47,48].

### 2.2. Depression

The Zung Self-Rating Depression Scale (SDS) [49] was used to measure depression symptomatology. The SDS includes ten positively worded and ten negatively worded questions which have been developed from factor analytic studies of MDD [1]. Responses indicate the frequency of each of the 20 SDS items for how often they have been experienced during the last two weeks: “None or a little of the time” (score = 1), “Some of the time” (2), “Good part of the time” (3), or “Most or all of the time” (4). SDS total raw scores range from 20 to 80 [49,50], and the SDS author defined raw scores of 40 or above as indicative of “clinically significant depression” [50]. The SDS has split-half reliability of 0.81 [49], 0.79 [51] and 0.94 [52], with an internal consistency (alpha) of 0.88 for depressed patients and 0.93 for non-depressed patients [53]. Because the four depression subtypes described by these data were derived from the SDS items, they are henceforth referred to as SDS subtypes for precision. Mean scores on each of the four SDS subtypes were calculated using the method described by Sharpley and Bitsika (2013) by summing the scores for SDS items as shown in the top section of Table 1, and then calculating the mean score so that differences in the total number of SDS items in each SDS subtype was not a confound [9].

### 2.3. EEG Data

EEG data were collected from 24 sites: Frontal lobe electrodes (FP1, FP2, F3, F4, F7, F8, FT7, FT8, FC3, FC4), Temporal lobe electrodes (T7, T8, TP7, TP8, C3, C4), Parietal lobe electrodes (P3, P4, P7, P8, CP3, CP4) and Occipital lobe electrodes (O1, O2) via a 40-channel Digital EEG Amplifier (NuAmps), using a *Quick Cap* with electrodes. Data were collected during continuous EEG measurement of 3 min Eyes Closed resting condition. EEG sites were cleaned with *Nuprep* gel, plus an alcohol swab before fitting the cap, and all electrode impedances were <5 KΩ. Participants were seated in an experimental booth while their EEG data were collected with a *Neuroscan* amplifier and a desktop computer (Compumedics, Abbotsford, VIC, Australia). EEG signals were recorded using the *Curry 7* software, at a sampling rate of 1 KHz, with the frequency band set using low and high filters of 8 Hz to 12.9 Hz (i.e., alpha band) and 13 to 18 Hz (beta band).

Data were processed using a low filter (high pass), a frequency of 1 Hz, and a slope of 2 Hz. In addition, a high filter (low pass) with frequency of 30 Hz and a slope of 8 Hz was applied. A notch filter of 50 Hz (Harmonics) with a slope of 1.5 Hz was used, plus a band stop filter with a frequency of 50 Hz (Harmonics) and a width of 10 Hz and slope of 5 Hz. A Hann window (with a 10% width to prevent data loss) was used to filter the data. Data were visually examined to identify artefacts (eye movements, muscle movements, spontaneous discharges or electrode pops, etc.), all of which were then removed from the data record. The magnitude of eye blink deflections was used to detect bad blocks by three automated methods (Subtraction, Covariance and Principal Component Analysis) to produce clean EEG data.

Back-to-back epochs of 4 secs duration were then created from the cleaned EEG data. Epochs were rejected if they still included clear visual evidence of artefacts listed above, or if an amplitude threshold of +/− 50 µV was breached after the epoch was baseline corrected (entire epoch used as baseline period). Most participants had over 90% usable artefact-free epochs for the Eyes Closed condition, whilst those with less than 75% artefact free epochs were excluded. This resulted in one participant being removed from the dataset. Spectral analysis was performed on these data for each participant using a Fast Fourier Transformation (FFT) to calculate the power spectra. The power values obtained from FFT were averaged across the 4 s EEG epochs. From this process, the total power within the alpha (8–12.9 Hz) and beta (13–18 Hz) frequency ranges was obtained for each EEG site for the Eyes Closed condition for each participant. These data were extracted and transferred to MATLAB 2022 and EEGLAB 2022 for visual analysis of comparative alpha and beta activity.

Connectivity was calculated for those networks that are most likely to be involved with thoughts, and emotions, rather than (for example) sensory integration or motor control. For example, the default mode network (DMN) has been recently found to exhibit inter- and intra-region connectivities that are associated with MDD [54,55]. Similarly, the Executive Control Network (ECN) has also been repeatedly described as interacting with depression severity [56,57], as has the Salience Network (SAL) [58,59]. These three major networks were selected for investigation as possible sources of difference between depressed and non-depressed participants within each of the four SDS subtypes.

Functional lagged linear connectivity (also known as. coherence) estimates of EEG frequency band activity were obtained in the alpha and beta bands for each available epoch using The Key Institute eLORETA (exact low resolution brain electromagnetic tomography) [60] software 20140711. eLORETA was chosen to measure functional connectivity as it allows cortical regions of interest to be used as nodes in the network analysis rather than electrode sites at the scalp. Combined with MRI data that has identified the locations of nodes for the DMN, SAL and ECN in the cortex [61], using eLORETA to calculate connectivity measures allows a more direct assessment of cortical activity based on current source estimates at the actual sites of each neural network, rather than attempting to estimate neural network interactions based on scalp electrode readings.

This technique provides a single weighted minimum norm solution to the inverse problem and has been demonstrated to provide zero error (but low spatial resolution) in localizing cortical grey matter test sources [62,63]. The weights utilized by eLORETA (based on the EEG montage used in the recording) are used to calculate current source density throughout the grey matter of a standardized realistic head model [64] based on the MNI152 template [65]. The resulting current density distribution is used to calculate measures of linear dependence between “virtual” electrodes placed at regions of interest (ROIs) within the grey matter; for a comprehensive description of the mathematics underlying the eLORETA methodology and how the weighted norms are calculated, see Pascual-Marqui [60,66]. ROIs were selected using commonly identified grey matter nodes in the DMN, SAL and ECN based on MNI coordinates as identified by Raichle (2011), with all grey matter tissue within 10 mm of the identified source included as part of that node [61]. This resulted in 18 ROIs being selected (see Table 2). In Table 2, ‘Location’ refers to the general identified brain region but MNI coordinates locate each region more specifically. MNI X describes the brain region according to its location from left (negative values) to right (positive values); MNI Y coordinates refer to the front (positive values) to rear (negative values); and TMNI Z coordinates position a brain site on the transverse plane of the brain, from top (positive values) to the bottom (negative values). Some regions listed in Table 2 are widespread, and thus occur in more than one network. In those cases, the regions may be more precisely identified by their MNI coordinates.

### 2.4. Procedure

Participants read an Explanatory Statement and signed a Consent Form, a background questionnaire (age, sex) and the SDS. Participants’ scalps were then prepared and the electrode cap fitted. Headphones were placed on participants so as to minimize the effect of external stimuli. Following 15 min of sitting still (adaptation), the audio-recorded experimental protocol (3 min Eyes Closed) was presented via headphones to ensure consistency across participants. Ethics approval was received from the Human Research Ethics Committee of the University of New England, Australia (Approval No. HE14-051; 25 March 2014) and all participants gave their written consent.

### 2.5. Data Analysis

Internal consistency for the SDS was determined via Cronbach’s alpha. Although some studies normalize EEG data, that process is likely to confound the results of statistical procedures [67]. Therefore, Spearman’s correlations were used to determine any significant associations between sex or age and SDS scores and EEG data, thus overcoming any confounds arising from parametric analyses with non-normal data.

The comparison between depressed and non-depressed participants’ coherence values for each SDS subtype was performed by independent samples *t*-tests. Because of the large number of associations measured, and the need to balance the consequent risk of a Type I error (inflated by family-wise error) versus a Type II error (by over-correction to the *p* value), the effect size and *p* value were adopted rather than the *p* value alone when considering results. In an exploratory study such as this, even small effects are valuable [68], and so Cohen’s d of at least 0.2 (corresponding to a *t* value of at least 2.0) was accepted as indicative of at least a difference suggestive of a meaningful effect between depressed and non-depressed participants’ SDS subtype scores that may warrant further investigation; *p* values are reported as adjunct data to support the d results.

This study is focused on a comparison of connectivity between the four SDS subtype scores, and therefore each instance of connectivity was cross-checked across all four SDS subtypes so that only those instances of connectivity that were exclusive to each SDS subtype were included in the comparison exercise.

## 3. Results

### 3.1. Age, Sex, SDS Scores

Internal consistency for the SDS (Cronbach’s alpha) was 0.905, and inspection of the Normal Q-Q plots for the SDS revealed an almost completely straight line, suggestive of normality. The mean SDS score was 36.70 (SD = 11.25), ranging from 21 to 66, with 67 participants having SDS scores lower than the cutoff for clinically significant depression [50], and 33 participants falling into the category of clinically significant depression as defined by Zung (1973) for the entire SDS. These two groups had significantly different total SDS scores (*M* SDS total score = 50.39, SD = 7.43 vs. M SDS total score = 29.95, SD = 4.83: *F*(1,99) = 273.729, *p* < 0.001, η_p_^2^ = 0.736) [50]. There were no significant correlations between age or sex and: SDS total score, any of the four SDS subtype scores, or any of the 24 EEG sites’ alpha or beta data.

### 3.2. MDD Subtypes

Mean scores were calculated for each of the four SDS subtypes described by Sharpley and Bitsika (2013) to overcome any confound that might arise from using total scores where different subtypes were measured by differing numbers of SDS items (as shown in Table 1, upper section) [9]. These mean scores were: Depressed mood = 1.736 (SD = 0.653; range = 1.00 to 3.67), Anhedonia = 1.850 (SD = 0.657; range = 1.00 to 3.75), Cognitive depression = 2.183 (SD = 0.810; range = 1.00 to 4.00), Somatic depression = 1.618 (SD= 0.511; range = 1.00 to 3.50), where 1 = “None or a little of the time”, 2 = “Some of the time”, 3 = “Good part of the time”, and 4 = “Most or all of the time”. That is, based upon Zung’s (1973) definition of clinically significant depression being a total score of 40/80 on the SDS, i.e., 50% of the possible total score, most of the four SDS subtypes’ mean scores were very close to that criterion [50]. Therefore, these means were used as cutoffs to identify the most depressed from the least depressed participants on each SDS subtype (defined as depressed versus non-depressed). All of the four SDS subtype scores were significantly correlated with each other (Table 1, lower section), accounting for between nearly 40% of the variance (between Anhedonia and Somatic depression) and over 86% of the variance (between Depressed mood and Cognitive depression). Although distinctly differentiated by clinical symptoms and underlying neurobiological pathways [9], these four SDS subtypes are (as expected) clearly related because they measure aspects of the same overall construct (i.e., depression).

Figure 1 shows the mean and standard error scores on their specific SDS subtype for each of the eight depression subgroups created by this process, and demonstrates their separation into depressed and non-depressed subgroups according to their severity on these SDS subtype-based metrics.

Differences in alpha and beta connectivity between depressed and non-depressed participants across four SDS subtypes.

#### 3.2.1. Number of Connections

The depressed and non-depressed participant subgroups that were identified according to their scores on the four SDS subtypes (described above), were then compared within each SDS subtype for the presence of any differences of at least a small effect size of d = 0.2 [69] in their connectivity values. From these data, only those connections that were exclusive to a particular SDS subtype were identified; Figure 2A–D show these data for alpha and beta bands. Using the presence of an effect size of at least d = 0.2, red lines indicate that the connectivity was less for the depressed participants than for the non-depressed participants; blue lines indicate that depressed participants’ connectivity was greater than that for the non-depressed participants. Each of the four SDS subtypes is shown separately, with the alpha data shown on the upper brain view panels, and the beta data shown on the lower two brain view panels for each subtype.

Figure 2A–D. Exclusive connections for four MDD subtypes, left, top, rear views.

#### 3.2.2. Networks Connected

Table 3 summarizes the number of exclusive connections shown in Figure 2A–D. Figure 3A–H provide further information in matrix format about which networks were connected, and in which direction. By reference to the *p* values provided by eLORETA in the bottom line of Table 3, it appears that only one (*p* = 0.048) reaches the traditional level of significance, with another indicating a trend at *p* = 0.059 in terms of the traditional Type I error rate. However, it must be acknowledged that the nonparametric permutation testing method used by eLORETA to determine significance thresholds consistently produces highly conservative threshold values [70,71]. Thus, although eLORETA brings a very conservative perspective to the issue of multiple comparisons, it is worth noting that there is some conjecture in the literature as to whether correction of *p* values is necessary [17], with some commentators arguing that is not (Feise, 2002; Rothman, 1990) [70,71].

To clarify the effect of eLORETA’s ultraconservative testing system, the alpha-band finding for Mood reported in Table 3 yielded a *t*-statistic of 4.003 and *p* = 0.048, but a typical analysis with the same sample size (100) would produce a *p*-value of 0.000134. By comparison, the weakest eLORETA-provided *p* value in Table 3 was for anhedonia in the alpha-band (*p* = 0.369). Typical analysis provided a *t* value of 2.725, *p* = 0.00782. Given the very conservative *p*-values that eLORETA provides, we are confident that using an eLORETA threshold of *p* > 0.05 does not automatically exclude some of these results from being of value in exploring the alpha- and beta-band signatures of these four MDD subtypes.

The first difference between the SDS subtypes’ connectivities depicted in Figure 2 and Figure 3 and Table 3 is that, by reference to the red and blue connectivity lines (Figure 2) and the red and blue squares (Figure 3), and also in Table 3, it is apparent that the *incidence* of alpha and beta band differences varied across the four SDS subtypes. For example, the most common occurrence was in Depressed mood and Somatic depression, with relatively fewer for Anhedonia and Cognitive depression. Second, the *direction* of those differences varied between subtypes, so that all the d ≥ 0.2 differences in the alpha band for Depressed mood were in the depressed < non-depressed direction, but for Cognitive depression the meaningful differences were in the depressed > non-depressed direction. Somatic depression showed a combination of depressed < non- depressed, and depressed > non-depressed differences; there were no meaningful alpha band differences between depressed and non-depressed participants for Anhedonia. In the beta band, Depressed mood and Somatic depression showed only meaningful depressed < non-depressed differences; Anhedonia exhibited meaningful differences in both directions; but Cognitive depression showed only meaningful differences in the depressed > non-depressed direction.

## 4. Discussion

These results are not unequivocal. Based on the eLORETA *p* values, mostly, they do not meet the usual standards required to determine an effect that is robust to Type I error. However, eLORETA *p* values are open to criticism of being ultraconservative, and comparison with typical *t*-tests indicated that the *p* values obtained and reported in Table 3 are worthy of consideration in this exploratory study. The d values also provide some argument for their consideration as suggestive of minor effects. More importantly, in the search for ways of defining the components of MDD, these results clearly represent a challenge to acceptance of the unitary model of depression, and provide some initial support for the four MDD subtypes investigated here. They also provide support for investigating the ways that the brain’s electrical activity might vary according to the various symptoms included in these four MDD subtypes. The primary aim of this study was to investigate and describe any differences in the alpha- and beta-band network connectivities between four MDD subtypes, defined by scores on groups of items from the SDS obtained by participants who met the criteria for clinically significant depression versus those who did not meet that criteria on each of the four SDS subtypes, based upon the model of clinically significant depression described for the entire SDS [50] as applied to participants’ scores for each SDS subtype. On that basis, depressed participants were distinguished from non-depressed participants on the bases of the severity of their scores on each SDS subtype (Figure 1), confirming the division of participants into these subgroups.

Alpha- and beta-bands were chosen for investigation because they indicate relatively different brain states (alpha = relaxed, beta = intense brain activity). Functional network connectivity was examined because of its ability to suggest how different regions of the brain may be involved in different cognitive, emotional, and behavioural activities. A secondary objective was to explore the possibility of depression circuits for each of the four SDS subtypes in the same way that this is being attempted for MDD [26,27]. As in most previous studies of the association between functional connectivity and depression [24], resting state EEG data were used in this study. As mentioned in the Introduction, this field is at the exploratory stage, and therefore no directional hypotheses could be generated for testing. Some subtype-specific findings are worthy of brief discussion.

*Depressed Mood*: Alpha connectivity pairs were identified within and between all three networks, indicating that mood-based depression may be a more global connectivity issue rather than due to activity within specific neural circuits. That said, the majority of connectivity differences were found in the SAL, indicating that mood-based depression may be a product of lower alpha power resulting from bottom-up processing from the limbic system [39,72]. Beta connectivity results suggested the dorsal ACC and the left anterior PFC within the SAL, and the right anterior PFC (SAL) and the dorsomedial PFC and the more ventral section of the right anterior PFC (both ECN), could be examined with some degree of optimism. Given the lateral anterior PFC’s role in cognitive and emotional control [73,74] and the dorsomedial PFC’s association with intention and decision-making [75], this could indicate that participants with low scores for depressed mood are more active in maintaining emotional and cognitive equilibrium than their more depressed peers.

*Anhedonia*: No differences were found in alpha connectivity between the depressed and non-depressed groups, but the depressed group showed greater beta connectivity in the left hemisphere, with connections from the left lateral parietal lobule (SAL) to the left superior parietal lobule (ECN) and a more frontal-posterior section of the left lateral parietal lobule (DMN). The increase in beta activity in regions associated with apathy [16,76,77] matches symptoms associated with anhedonic depression. Greater beta connectivity in the non-depressed group was almost exclusively in the right hemisphere, with the focus of connectivity at the right anterior PFC (ECN) with multiple connections to both the DMN and SAL. While several studies have associated anhedonic states with primarily right-hemisphere activity (e.g., Bruder et al., 1998; Shaw et al., 2021 [78,79]) this small network may reflect the emotional control necessary to keep such states in check, given the right anterior PFC’s role in regulating emotion [73,74].

*Cognitive depression*: Alpha connectivity results showed a trend towards greater connectivity involving the right inferior temporal lobe, a region of the DMN typically involved in task-related visual processing [80]. This may reflect the tendency of those with depression to have a relative lack of interest in their surroundings.

Beta connectivity indicated a single link between the dorsomedial PFC and the left anterior prefrontal cortex. This section of the dorsomedial PFC is associated with intention and decision-making as part of the ECN [75], while the left anterior prefrontal cortex is associated with emotional control as part of the SAL [81]. Taken together, these functions indicate that participants with low scores on Cognitive depression were more active in making choices in order to maintain an appropriate mental state.

*Somatic depression*: Increased alpha connectivity was found for both the depressed and non-depressed samples. The depressed sample showed two connectivity pairs from the right lateral parietal lobule (DMN), which has often been linked with increased apathy in a range of disorders [16,76,77].

Conversely, the non-depressed group showed two distinct neural signatures in alpha connectivity. Firstly: a frontal set of three connections all involving the dorsomedial PFC (ECN); these include the right anterior PFC (ECN), the left anterior PFC (SAL), and the right insula (SAL). This network of PFC nodes is known to be involved in emotional control and decision-making [75,81]. However, the addition of the insula and its role in maintaining awareness of emotional states [82,83] may be indicative of non-depressed participants’ comparatively lower anxiety compared to those with somatic depression. The second signature, a more posterior set of connections primarily involving the left insula (SAL) and the left lateral parietal lobule (DMN), may reflect the appropriate level of relaxation or need for intervention in controlling emotional states in those without somatic depression.

Beta connectivity differences in somatic depression were limited to a single connection between the dorsomedial PFC (ECN) and the dorsal ACC (SAL), which was greater for the non-depressed group. Given the role of the dorsomedial PFC in intention and decision-making [75], plus the dorsal ACC’s role in maintaining motivation and emotional stability [84,85], an increase in beta connectivity in the depressed group may reflect the restless and anxiety-related symptoms in those with somatic depression.

In terms of clinical implications of these initial findings, it must be emphasized that further research is necessary to provide a firmer basis for therapy, but initial suggestions could be towards identification of neural circuits specific to particular MDD subtypes, plus the effects of interventions such as TMS, medication, neurofeedback aimed at influencing the connectivity variables found here. Further research would also enable the possible use of these connectivity findings as potential biomarkers of MDD subtypes.

As mentioned above, most of these results are open to challenge on the basis of the very conservative eLORETA *p* values, and must be considered as preliminary at this stage. As such, caution must be exerted when attempting to form a solid model of four robust subtypes of depression solely based upon alpha and beta connectivity differences. However, these results hint to possible differences that may be revealed by further investigation of EEG-related variables. For example, although an argument was made in the Introduction for examining alpha- and beta-wave activity because of the comparatively different kind of neurocognitive activity that these wavelengths represent, those are not the only possibly relevant wavelengths of brain electrical activity that could be investigated to describe the neurological underpinnings of MDD subtypes. Gamma-wave activity represents maximum concentration and cognitive effort, of the kind undertaken when attempting to process complex information, which may occur when an individual is forced to deal with aversive emotions (e.g., Depressed mood) but which may not be so prevalent when experiencing the symptoms of Somatic depression. Thus, extension of this research into other wavelengths of brain activity has the potential to further describe how different MDD symptoms (i.e., subtypes) might initiate different forms of electrical activity in various regions of the brain.

Other limitations restrict the internal and external validity of these exploratory findings. For example, sample size, the voluntary nature of the participants, the lack of a clearly identified group of patients suffering from clinically interviewed MDD, the use of self-reports of depressive symptomatology, and the collection of data at a single time point. Although the use of a relatively small effect size (d = 0.2) here was justified by the exploratory nature of this study, it does not represent the standard needed to form solid conclusions, and must remain indicative only at this stage. Further, the restricted sample size in this exploratory study prevented the valid use of confidence intervals for the effect sizes, which would have been necessarily extended simply due to the sample size. Replication with larger samples that provide greater statistical power is necessary. Stress is associated with depression in a causal chain [86], and collection of data across a longer time period would enable comparisons to be made regarding any changes in the EEG-SDS subtype results found here. The choice of which four MDD subtypes to use was based upon the previous literature [9], but there are other depression subtypes [1,5,87], and an investigation of the EEG profiles of these would extend the current findings and facilitate a wider understanding of the heterogeneity of depression [42,88]. The methodological approach used in this study may be seen as *a priori* because MDD subtypes were defined before exploration of EEG data, and was adopted because of its close resemblance to clinical practice (when patients present with specific MDD symptoms), but it must be acknowledged that use of an *a posteriori* model of collecting EEG data first and then regressing back to groups of MDD symptoms also holds value. Which of these two approaches is most valid, or valuable, is yet to be determined, but they are both useful in developing a greater understanding of the neurological underpinnings of MDD, and hence developing more appropriate treatment regimes. It should be noted that eLORETA produces ultra-conservative *p* values, and therefore some of the associations discussed above, e.g., anhedonia) should be considered to be exploratory only. It must also be noted that although this study provides meaningful correlations across brain regions for individual subtypes of depression, additional work is required to validate these. The addition of fMRI or tractography would serve as powerful cross-validation, though to date, no such data have been collected on patients clinically stratified at the same level of accuracy as in this study. In line with the goal of this study as an exploratory pilot, the correlations presented here should be interpreted as suggestive. Future studies are also required to account for the effects of medications, sleep deprivation, anxiety levels, disease stability and other relevant environmental factors.

Finally, although EEG data are well-established as representations of neurocognitive activity, other methodologies for investigating brain activity (e.g., fMRI) remain potentially valuable sources of further information about the nature of MDD subtypes, as mentioned in the Introduction. Application of these to the current study aims would enable direct comparisons between the present results and others, which was relatively difficult due to the lack of relevant previous studies.

## 5. Conclusions

Accepting these limitations, and the very initial nature of the findings, these results identify possible future directions for research aimed at exploration of the ways in which subtypes of depression may differ neurologically. As such, they are congruent with suggestions of MDD being heterogenous rather than unitary [2,43], and underlie the need for consideration of this variability in presenting symptomatology in depressed patients.

## Figures and Tables

**Figure 1 jcm-14-05295-f001:**
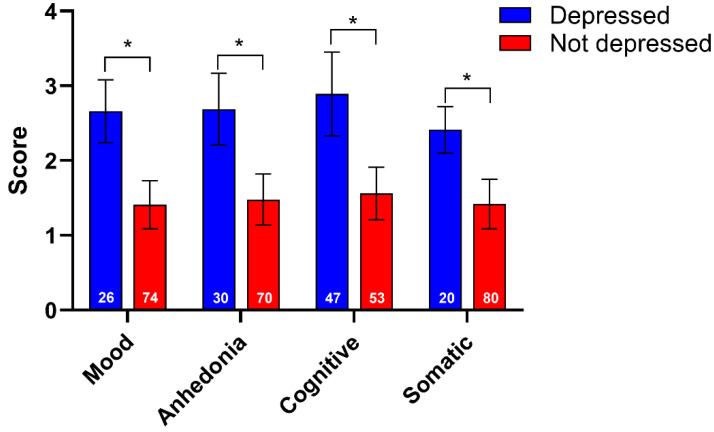
Mean (SE) scores for depressed vs. not-depressed participants within each SDS subtype (subgroup sizes shown in lower part of columns). * *p* < 0.001.

**Figure 2 jcm-14-05295-f002:**
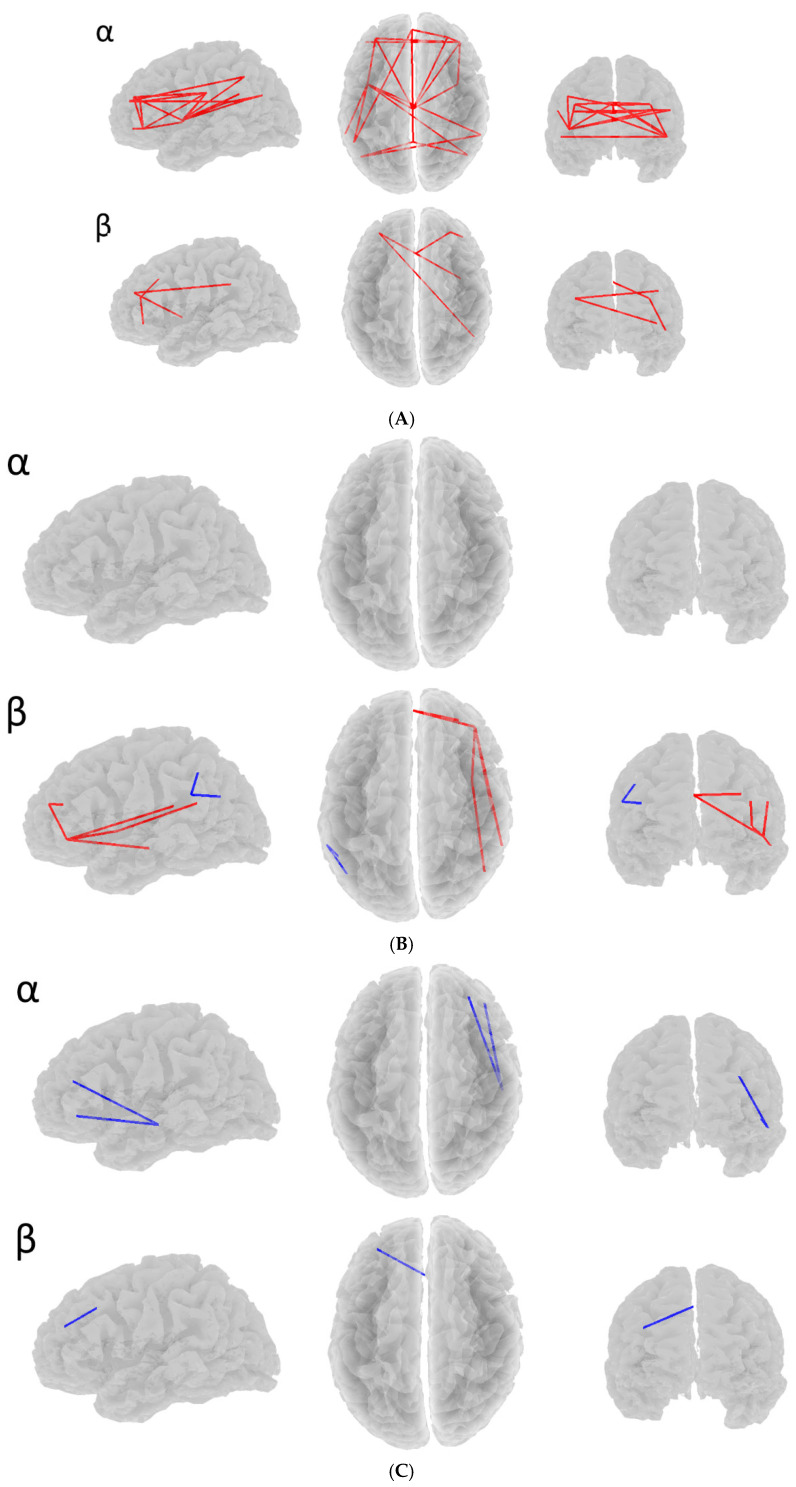
(**A**). Depressed mood: Exclusive connections for Depressed versus Non-depressed participants. Alpha (**upper view**) and beta (**lower view**). Red lines indicate d ≥ 0.2 greater connectivity for non-depressed participants than depressed participants. (**B**). Anhedonia: Exclusive connections for Depressed versus Non-depressed participants. Alpha (**upper view**) and beta (**lower view**). Red lines indicate d ≥ 0.2 greater connectivity for non-depressed participants than depressed participants; blue lines indicate greater connectivity for depressed than non-depressed participants. (**C**). Cognitive depression: Exclusive connections for Depressed versus Non-depressed participants. Alpha (**upper view**) and beta (**lower view**). Blue lines indicate greater connectivity for depressed than non-depressed participants. (**D**). Somatic depression: Exclusive connections for Depressed versus Non-depressed participants. Alpha (**upper view**) and beta (**lower view**). Red lines indicate d ≥ 0.2 greater connectivity for non-depressed participants than depressed participants; blue lines indicate greater connectivity for depressed than non-depressed participants.

**Figure 3 jcm-14-05295-f003:**
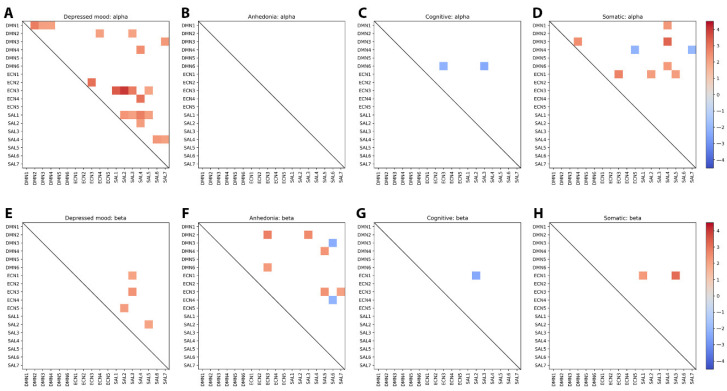
(**A**–**H**). Exclusive network connection differences between Depressed vs. Non-depressed participants (effect size of d = 0.2) [69] across four SDS subtypes ^1^. ^1^ Red = Depressed < non-depressed; blue = Depressed > non-depressed.

**Table 1 jcm-14-05295-t001:** Four depression subtypes and relevant Zung SDS ^1^ items (from Sharpley and Bitsika, 2013) [9].

Subtype	Depressed Mood	Anhedonia		Cognitive Depression		Somatic Depression
SDS items	1. I feel downhearted and blue 3. I have crying spells or feel like it 14. I feel hopeful about the future 15. I am more irritable than usual 17. I feel that I am useful and needed 19. I feel that others would be better off if I were dead	5. I eat as much as I used to 6. I still enjoy sex 18. My life is pretty full 20. I still enjoy doing the things I used to		11. My mind is as clear as it used to be 12. I find it easy to do the things I used to do 16. I find it easy to make decisions		4. I have trouble sleeping at night 7. I notice that I am losing weight 8. I have trouble with constipation 9. My heart beats faster than usual 10. I get tired for no reason 13. I am restless and can’t keep still
Subtype	Inter-subtype correlation coefficients					
	Anhedonia		Cognitive depression		Somatic depression	
Depressed mood	0.678 *		0.831 *		0.748 *	
Anhedonia			0.789 *		0.626 *	
Cognitive depression					0.736 *	

^1^ Self-reported Depression Scale; * *p* < 0.001.

**Table 2 jcm-14-05295-t002:** Networks, brain sites and MNI ^1^ coordinates of the 18 Regions of Interest selected from study.

Network	Location	MNI X	MNI Y	MNI Z
DMN1 ^2^	Posterior cingulate	0	−52	27
DMN2	Medial PFC	−1	54	27
DMN3	L lateral parietal	−46	−66	30
DMN4	R lateral parietal	49	−63	33
DMN5	L inferior temporal	−61	−24	−9
DMN6	R inferior temporal	58	−24	−9
ECN1 ^3^	Dorsal medial PFC	0	24	46
ECN2	L anterior PFC	−44	45	0
ECN3	R anterior PFC	44	45	0
ECN4	L superior parietal	−50	−51	45
ECN5	R superior parietal	50	−51	45
SAL1 ^4^	Dorsal ACC	0	−21	36
SAL2	L anterior PFC	−35	45	30
SAL3	R anterior PFC	32	45	30
SAL4	L insula	−41	3	6
SAL5	R insula	41	3	6
SAL6	L lateral parietal	−62	−45	30
SAL7	R lateral parietal	62	−45	30

^1^ Montreal Neurological Institute system; ^2^ Default Mode Network; ^3^ Executive Control Network; ^4^ Salience Network, Dorsal Anterior Cingulate Cortex.

**Table 3 jcm-14-05295-t003:** Number of significant (d ≥ 0.2) connections for depressed versus non-depressed participants in the alpha and beta bands.

SDS Subtype	Dm ^1^	Dm	Da	Da	Dc	Dc	Ds	Ds
Band	α	β	α	β	α	β	α	β
D > ND ^2^	0	0	0	2	2	1	2	0
D < ND	20	4	14	6	0	0	6	2
Best *p* from eLORETA	0.048	0.186	0.369	0.130	0.245	0.228	0.059	0.222

^1^ Depressed mood; ^2^ Depressed > non-depressed.

## Data Availability

Data are available on request from the author.

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
