# Peer review of "An Exploratory Comparison of Alpha and Beta Network Connectivity Across Four Depression Subtypes"

_jcm, 2025, doi:10.3390/jcm14155295_

Round 1
Reviewer 1 Report
Comments and Suggestions for Authors
This article explores EEG-based alpha and beta network connectivity in 4 subtypes of MDD: Depressed Mood, Anhedonia, Cognitive Depression, and Somatic Depression. By comparing depressed and non-depressed individuals within each subtype, it identifies unique connectivity patterns that could represent distinct neurological signatures. This research is significant as it challenges the traditional unitary model of MDD and contributes to more personalized diagnostic and therapeutic approaches in clinical psychiatry.
Unexplained Acronyms
- The acronym FFT first appears in Section 2.2 (“EEG Preprocessing”) in the phrase “...the time series data was converted to the frequency domain using a Fast Fourier Transformation.” While the full term is provided, the abbreviation (FFT) is not included at that point, despite the acronym being used subsequently (e.g., “FFT-transformed”). the first that the acronym is used is should be explained
- Similarly, the acronym ACC first appears in Table 2 (Section 2.3, “Brain Networks and Electrodes”) as “Dorsal ACC”, but the full term anterior cingulate cortex is never introduced before or within the table. This violates the requirement that all acronyms be introduced with their full form followed by the abbreviation in parentheses upon first use..
Introduction
- Include a brief sentence explicitly summarizing the study’s objectives and hypotheses or expected.
Materials and Methods
- SDS-derived subtypes are justified but provide a clearer explanation or citation validating their construct validity beyond the authors’ own prior work.
- EEG Preprocessing is examined with Excellent detail, but the role of artifact rejection and its thresholds should better quantified (e.g., percentage of data excluded).
- Please include a brief clarification on why eLORETA was chosen over other alternatives Functional lagged linear connectivity is suitable
Results
- The manuscript consistently omits the leading zero before decimal points in the body text—for example, “Internal consistency for the SDS (Cronbach’s alpha) was .905.” Authors should including a zero before decimal points (e.g., 0.905 instead of .905) in the main text and tables.
- When Effect sizes are used (d = 0.2), authors should add confidence if available.
- Statistical significance should not be the sole criterion for interpreting results because overemphasis on p-values can obscure meaningful patterns. Nevertheless s
- some findings in the manuscript with clearly non-significant p-values (e.g., p > 0.05) are discussed with the same weight as statistically supported effects. We recommend that the authors clearly indicate when such findings are preliminary or hypothesis-generating. For example, in the discussion of anhedonia, beta connectivity differences (e.g., p = 0.130 and p = 0.369) are interpreted functionally without clear mention of their exploratory status
Discussion
- Participants were not clinically diagnosed with MDD via interview, limiting generalizability. This should be commented as a limitation..
- We recommend that the authors expand the discussion to address how the current findings could inform future studies focused on therapeutic applications. Specifically, the distinct patterns of alpha- and beta-band connectivity observed across MDD subtypes may guide future research aimed at identifying subtype-specific neural circuits as potential treatment targets. Subsequent studies could explore whether interventions such as transcranial magnetic stimulation (TMS), neurofeedback, or pharmacological modulation can be tailored to alter connectivity within specific networks (e.g., DMN, ECN, SAL) relevant to each subtype. Additionally, future research might investigate the utility of these EEG connectivity patterns as biomarkers to support more personalized diagnosis and treatment monitoring in depression. Including these perspectives would underscore the clinical potential of the findings and help frame directions for translational research.
Author Response
Reviewer 1
This article explores EEG-based alpha and beta network connectivity in 4 subtypes of MDD: Depressed Mood, Anhedonia, Cognitive Depression, and Somatic Depression. By comparing depressed and non-depressed individuals within each subtype, it identifies unique connectivity patterns that could represent distinct neurological signatures. This research is significant as it challenges the traditional unitary model of MDD and contributes to more personalized diagnostic and therapeutic approaches in clinical psychiatry.
Unexplained Acronyms
- The acronym FFT first appears in Section 2.2 (“EEG Preprocessing”) in the phrase “...the time series data was converted to the frequency domain using a Fast Fourier Transformation.” While the full term is provided, the abbreviation (FFT) is not included at that point, despite the acronym being used subsequently (e.g., “FFT-transformed”). the first that the acronym is used is should be explained
Author response: We think that the reviewer must be thinking of another ms, because section 2.3 in this ms is “Depression”. The acronym “FFT” first appears on lines 241-2, and is defined there.
- Similarly, the acronym ACC first appears in Table 2 (Section 2.3, “Brain Networks and Electrodes”) as “Dorsal ACC”, but the full term anterior cingulate cortex is never introduced before or within the table. This violates the requirement that all acronyms be introduced with their full form followed by the abbreviation in parentheses upon first use..
Author response: We have now added a footnote to Table 2 to define ACC.
Introduction
- Include a brief sentence explicitly summarizing the study’s objectives and hypotheses or expected.
Author response: We have now divided section 1 into subsections, one of which (1.6) is titled “Study aims”.
Materials and Methods
- SDS-derived subtypes are justified but provide a clearer explanation or citation validating their construct validity beyond the authors’ own prior work.
Author response: We have now added several references to lines 55-6 confirming the validity of the four MDD subtypes.
- EEG Preprocessing is examined with Excellent detail, but the role of artifact rejection and its thresholds should better quantified (e.g., percentage of data excluded).
Author response: We have added the following at lines 242-246: “Epochs were rejected if they still included clear visual evidence of artefacts listed above, or if an amplitude threshold of +/- 50 µV was breached after the epoch was baseline corrected (entire epoch used as baseline period). Most participants had over 90% usable artefact-free epochs for the Eyes Closed condition, whilst those with less than 75% artefact free epochs were excluded. This resulted in one participant being removed from the dataset.”
- Please include a brief clarification on why eLORETA was chosen over other alternatives Functional lagged linear connectivity is suitable
Author response: We have now added the following information at lines 266-72: “ eLORETA was chosen to measure functional connectivity as it allows cortical regions of interest to be used as nodes in the network analysis rather than electrode sites at the scalp. Combined with MRI data that has identified the locations of nodes for the DMN, SAL and ECN in the cortex (Raichle, 2011), using eLORETA to calculate connectivity measures allows a more direct assessment of cortical activity based on current source estimates at the actual sites of each neural network, rather than attempting to estimate neural network interactions based on scalp electrode readings”.
Results
- The manuscript consistently omits the leading zero before decimal points in the body text—for example, “Internal consistency for the SDS (Cronbach’s alpha) was .905.” Authors should including a zero before decimal points (e.g., 0.905 instead of .905) in the main text and tables.
Author response: Corrected.
- When Effect sizes are used (d = 0.2), authors should add confidence if available.
Author response: In general, this is true. However, there is an inverse relationship between sample size and CI, so that, when the sample size is restricted (as in this study), confidence intervals can be very wide, not necessarily reflecting the true situation that might become more apparent with a larger sample. In acknowledgement of this limitation, we have now added the following statement in the Limitations lines 566-569: “Further, the restricted sample size in this exploratory study prevented the valid use of confidence intervals for the effect sizes, which would have been necessarily extended simply due to the sample size. Replication with larger samples that provide greater statistical power is necessary.”
- Statistical significance should not be the sole criterion for interpreting results because overemphasis on p-values can obscure meaningful patterns. Nevertheless s
- some findings in the manuscript with clearly non-significant p-values (e.g., p > 0.05) are discussed with the same weight as statistically supported effects. We recommend that the authors clearly indicate when such findings are preliminary or hypothesis-generating. For example, in the discussion of anhedonia, beta connectivity differences (e.g., p = 0.130 and p = 0.369) are interpreted functionally without clear mention of their exploratory status
Author response: We did address this issue in lines 405-422, commenting on eLORETA’s ultra-conservative p values, and also in the first few lines of the Discussion (ie., “eLORETA p values are open to criticism of being ultraconservative, and comparison with typical t-tests indicated that the p values obtained and reported in Table 3 are worthy of consideration in this exploratory study”.
To further emphasise that point, we have added the following statement to the Limitations lines 584-594: “It should be noted that eLORETA produces ultra-conservative p values, and therefore some of the associations discussed above, e.g., anhedonia) should be considered to be exploratory only”.
Discussion
- Participants were not clinically diagnosed with MDD via interview, limiting generalizability. This should be commented as a limitation..
Author response: we have already addressed this limitation in our comment on lines 561-563 :”the lack of a clearly-identified group of patients suffering from clinically-interviewed MDD,”.
- We recommend that the authors expand the discussion to address how the current findings could inform future studies focused on therapeutic applications. Specifically, the distinct patterns of alpha- and beta-band connectivity observed across MDD subtypes may guide future research aimed at identifying subtype-specific neural circuits as potential treatment targets. Subsequent studies could explore whether interventions such as transcranial magnetic stimulation (TMS), neurofeedback, or pharmacological modulation can be tailored to alter connectivity within specific networks (e.g., DMN, ECN, SAL) relevant to each subtype. Additionally, future research might investigate the utility of these EEG connectivity patterns as biomarkers to support more personalized diagnosis and treatment monitoring in depression. Including these perspectives would underscore the clinical potential of the findings and help frame directions for translational research.
Author response: Agreed. We have added the following para to the Discussion (lines 537-542): “In terms of clinical implications of these initial findings, it must be emphasised that further research is necessary to provide a firmer basis for therapy, but initial suggestions could be towards identification of neural circuits specific to particular MDD subtypes, plus the effects of interventions such as TMS, medication, neurofeedback aimed at influencing the connectivity variables found here. Further research would also enable the possible use of these connectivity findings as potential biomarkers of MDD subtypes.”
Reviewer 2 Report
Comments and Suggestions for Authors
The manuscript presents a well-conducted study exploring EEG alpha- and beta-band connectivity patterns associated with four clinically defined MDD subtypes. Using resting-state EEG and source-space electrical analysis, the authors compare depressed and non-depressed participants on each subtype employing both frequentist (t-tests) and effect size measures. The study offers comprehensive content and insightful analysis; however, to further enhance clarity and the overall impact, the manuscript would benefit from revision addressing the following minor points:
- The introduction gives a good sense of the topic and research goals, but it would benefit from a clearer statement of what this study actually contributes to the field. Right now, it’s not easy to tell what’s new or different about this work compared to existing research. I’d suggest adding a short paragraph or even just a few sentences near the end of the introduction that directly state the key contributions.
- The discussion chapter provides a good description of the results and links them to anatomo-functional data, but it requires a more rigorous scientific dialogue with previous studies and clearer structuring of the discussion. There are no direct parallels or contrasts drawn between the current findings and specific literature data (for example, statements like “unlike Smith et al. (2019), we found…” or “our results are consistent with…” are missing). Additionally, there is no analysis of why the obtained results might differ from other studies, taking into account methodological, sample-related, or technical factors. While some limitations are partially acknowledged, it would be beneficial to more strongly emphasize the need for replication, the use of additional methods, and the expansion of the range of EEG frequency bands studied.
Author Response
The manuscript presents a well-conducted study exploring EEG alpha- and beta-band connectivity patterns associated with four clinically defined MDD subtypes. Using resting-state EEG and source-space electrical analysis, the authors compare depressed and non-depressed participants on each subtype employing both frequentist (t-tests) and effect size measures. The study offers comprehensive content and insightful analysis; however, to further enhance clarity and the overall impact, the manuscript would benefit from revision addressing the following minor points:
- The introduction gives a good sense of the topic and research goals, but it would benefit from a clearer statement of what this study actually contributes to the field. Right now, it’s not easy to tell what’s new or different about this work compared to existing research. I’d suggest adding a short paragraph or even just a few sentences near the end of the introduction that directly state the key contributions.
Author response: We have a large para at the end of the Introduction (section 1.6) that sets out the aims of the study, which we believe appropriately addresses the reviewer’s comment.
- The discussion chapter provides a good description of the results and links them to anatomo-functional data, but it requires a more rigorous scientific dialogue with previous studies and clearer structuring of the discussion. There are no direct parallels or contrasts drawn between the current findings and specific literature data (for example, statements like “unlike Smith et al. (2019), we found…” or “our results are consistent with…” are missing). Additionally, there is no analysis of why the obtained results might differ from other studies, taking into account methodological, sample-related, or technical factors. While some limitations are partially acknowledged, it would be beneficial to more strongly emphasize the need for replication, the use of additional methods, and the expansion of the range of EEG frequency bands studied.
Author response: We Have emphasized throughout the ms that this is an exploratory study, due to the lac of previous research on these four MDD subtypes. To emphasise this point, we have added the following statement to the last para of the Introduction (lines 170-3): “Due to the lack of previous studies of these four MDD subtypes and their comparative EEG data, it was not possible to set this study within the previous research. Instead, this study is necessarily exploratory, seeking to identify possible associations that might be used as hypotheses for future research.”
We agree with this comment that our study needs replication, and we have stated that several times in the para from lines 566-9. To add more of this would be repetitious.
Reviewer 3 Report
Comments and Suggestions for Authors
Good day,
This study has many scientific deficiencies and methodological problems. Assuming B frequency is associated with increase thinking and can be linked to depression is not proven and the studies cannot substantiate any relationships without correlating with functional MRI and tract images. It has suppositions such as correlation to MDD subtypes but no substantial clinical or scientific correlate to prove EEG relationships. The quoted studies also are not indicating a definitive correlate between MDD and EEG variances. Also you have to account for medications and the environment and stability of disease, sleep deprivation, anxiety levels, even in the normal population tested.
Suggest consider re-tooling the paper as a suggestive pilot initiative and list these limitaitons
Author Response
This study has many scientific deficiencies and methodological problems. Assuming B frequency is associated with increase thinking and can be linked to depression is not proven and the studies cannot substantiate any relationships without correlating with functional MRI and tract images. It has suppositions such as correlation to MDD subtypes but no substantial clinical or scientific correlate to prove EEG relationships. The quoted studies also are not indicating a definitive correlate between MDD and EEG variances. Also you have to account for medications and the environment and stability of disease, sleep deprivation, anxiety levels, even in the normal population tested.
Author response: We have added the following statement to the Limitations lines 586-94 to emphasise the exploratory nature of this study. With many other similar caveats throughout the ms, we believe that we have sufficiently alerted the reader to the very initial nature of these findings. “It must be noted that although this study provides meaningful correlations across brain regions for individual subtypes of depression, additional work is required to validate these. The addition of fMRI or tractography would serve as powerful cross-validation, though to date, no such data have been collected on patients clinically stratified at the same level of accuracy as in this study. In line with the goal of this study as an exploratory pilot, the correlations presented here should be interpreted as suggestive. Future studies are also required to account for the effects of medications, sleep deprivation, anxiety levels, disease stability and other relevant environmental factors.”
Round 2
Reviewer 3 Report
Comments and Suggestions for Authors
Limitations are reasonable and states the limitations of the study